# Surgical Strikes on Host Defenses: Role of the Viral Protease Activity in Innate Immune Antagonism

**DOI:** 10.3390/pathogens11050522

**Published:** 2022-04-28

**Authors:** Chue Vin Chin, Mohsan Saeed

**Affiliations:** 1Department of Biochemistry, Boston University School of Medicine, Boston, MA 02118, USA; chuevin@bu.edu; 2National Emerging Infectious Diseases Laboratories, Boston University, Boston, MA 02118, USA

**Keywords:** viral proteases, innate immunity, immune antagonism, interferon pathway, virus-induced proteolysis

## Abstract

As a frontline defense mechanism against viral infections, the innate immune system is the primary target of viral antagonism. A number of virulence factors encoded by viruses play roles in circumventing host defenses and augmenting viral replication. Among these factors are viral proteases, which are primarily responsible for maturation of viral proteins, but in addition cause proteolytic cleavage of cellular proteins involved in innate immune signaling. The study of these viral protease-mediated host cleavages has illuminated the intricacies of innate immune networks and yielded valuable insights into viral pathogenesis. In this review, we will provide a brief summary of how proteases of positive-strand RNA viruses, mainly from the *Picornaviridae*, *Flaviviridae* and *Coronaviridae* families, proteolytically process innate immune components and blunt their functions.

## 1. Introduction

Human cells are equipped to defend themselves against viral infections. When encountered by a virus, they mount a rapid and potent immune response that creates a protective environment in the infected cell and alerts the neighboring cells to an ongoing viral assault [1,2]. A key to the success of this defense mechanism is an early detection of viral components, such as nucleic acids and proteins, collectively called pathogen-associated molecular patterns (PAMPs) [3,4,5], and a quick relay of this information to immune effector molecules, which act as foot soldiers of the innate defense system and employ a variety of mechanisms to block viral propagation [1,6,7]. From detecting viral signatures to launching an effective counter-offensive against viral invasions, all steps in host cells are carried out by specialized proteins, some of which are constitutively expressed, while others are present at sub-detectable levels or exist in inactive forms during peacetime and only come into existence or action when the cell is under threat. A highly coordinated action of all of these proteins creates an intracellular environment averse to viral replication and spread.

Viruses, on the other hand, must circumvent host defenses to complete their lifecycle [8]. Therefore, it is no surprise that they have evolved a myriad of strategies to block innate immune signaling at almost every step along the way. These strategies include sequestration or degradation of host antiviral proteins [9,10,11,12,13,14], production of decoys to trick the immune system into chasing red herrings [15,16,17], shielding viral components from immune surveillance [16,18], and suppressing the expression of antiviral genes [16,19,20,21,22]. A number of viruses, particularly those with a positive-sense RNA genome, encode proteases, necessitated by the requirement to cleave polyproteins generated during the lifecycle of these viruses. A large body of literature shows that viral proteases are key virulence factors that contribute to viral pathogenesis by limiting host responses [10,23,24,25]. They do so by interacting with innate immune components in a non-catalytic fashion [9,26,27,28], proteolytically cleaving cellular proteins involved in innate immunity [10,14,25], and altering protein post-translational modifications critical for signal transduction [29,30,31,32,33]. This review focuses on proteolytic processing of host proteins and how it incapacitates the antiviral immune system. We begin by introducing proteases of three positive-strand RNA virus families, *Picornaviridae*, *Flaviviridae*, and *Coronaviridae*, followed by a brief description of key cellular proteins involved in innate immune signaling and their targeting by viral proteases.

## 2. Proteases of Positive-Strand RNA Viruses

Most positive-strand RNA viruses encode one or more proteases, which play key roles in maturation of viral proteins and inhibition of host antiviral functions. The role of the viral protease activity in host antagonism has been most extensively studied in the context of picornaviruses, flaviviruses, and coronaviruses. Picornaviruses are small non-enveloped viruses with a genome size between 7.5 and 10 kb and can be grouped into 68 genera (as of July 2021) [34]. The Enterovirus genus is the largest with several medically important members, such as poliovirus (PV), Coxsackieviruses (CVs), human rhinoviruses (HRVs), enterovirus D68 (EV-D68), EV-D70, and EV-A71. Other notable genera include Cardiovirus (containing encephalomyocarditis virus (EMCV)), Hepatovirus (containing hepatitis A virus (HAV)), and Aphthovirus (containing foot-and-mouth disease virus (FMDV)). The genome of these viruses encodes a single polyprotein of approximately 3000 amino acids length, which is processed by viral proteases into 11–12 mature proteins [35,36,37] (Figure 1A). The viral proteases involved in polyprotein processing include 2A^pro^ and 3C^pro^ for enteroviruses, L^pro^ and 3C^pro^ for aphthoviruses, and 3C^pro^ for hepatoviruses and cardioviruses [38,39]. The 2A^pro^ and 3C^pro^ are cysteine proteases that adopt the chymotrypsin-like fold [40,41,42,43], whereas L^pro^ adopts the papain-like fold [44,45].

The *Flaviviridae* family includes a number of clinically important viruses that mainly fall into two genera, Flavivirus and Hepacivirus. The various members of the genus Flavivirus include Zika virus (ZIKV), dengue virus (DENV), yellow fever virus (YF), West Nile virus (WNV), Japanese encephalitis virus (JEV), St. Louis encephalitis virus (SLEV), and tick-borne encephalitis virus (TBEV). These viruses are highly pathogenic to humans, mainly transmitted through mosquitoes and ticks, and prevalent in different parts of the world. The ~11 kb genome of flaviviruses encodes a polyprotein that is cleaved by host proteases and a single viral protease, NS2B-NS3^pro^, into 10 mature proteins [48,49] (Figure 1B). The NS3^pro^ functions as an active enzyme only in the presence of the cofactor NS2B, which is an integral membrane protein of 14 kDa [48,50,51]: The NS3 protein is insoluble and catalytically inactive in the absence of NS2B, suggesting that NS2B has a role in NS3 folding [52,53,54,55]. The genus Hepacivirus includes hepatitis C virus (HCV), which is responsible for an estimated 58 million cases of chronic hepatitis worldwide [56]. The 9.6 kb genome of this virus is translated into a single polyprotein, which is then processed by host and two viral proteases, NS2^pro^ and NS3-NS4A^pro^, to yield 10 mature proteins [57,58]. NS3-NS4A^pro^, the main protease responsible for polyprotein processing, forms a noncovalent heterodimer consisting of a catalytic subunit NS3 and an activating cofactor NS4A [59,60]. The NS2B-NS3^pro^ and NS3-NS4A^pro^ are serine proteases with the chymotrypsin-like fold [61,62,63].

The *Coronaviridae* family has recently gained global attention due to the unprecedented pandemic caused by the newly emerged severe acute respiratory syndrome coronavirus 2 (SARS-CoV-2) [64,65]. This virus was first identified in December 2019 [64], and as of April 2022, has infected approximately 500 million individuals around the world and claimed around 6 million lives [66]. Before this, two other coronaviruses, SARS-CoV [67,68] and Middle East respiratory syndrome coronavirus (MERS-CoV) [69], made a zoonotic jump into humans and triggered large-scale outbreaks of severe respiratory disease during the past two decades. In addition to these highly pathogenic viruses, the *Coronaviridae* family also contains endemic viruses, such as NL-63, OC-43, 229-E, and HKU-1, which exhibit seasonality and mostly cause mild respiratory disease [70]. Coronaviruses have the largest genome (26–32 kb) of all known RNA viruses. The 5′-terminal two-third of the viral genome contains two open-reading frames (ORFs), 1a and 1b. ORF1a codes for polyprotein 1a, whereas ORF1a and 1b together encode polyprotein 1ab [71]. This latter mechanism is mediated by a (−1) ribosomal frameshift overreading the stop codon of ORF1a [72,73]. The polyproteins 1a and 1ab are processed into 16 mature proteins by two viral proteases, papain-like protease (PL^pro^) and a 3C-like protease (3CL^pro^) [74,75] (Figure 1C). Both PL^pro^ and 3CL^pro^ are cysteine proteases, however, while PL^pro^ adopts the papain-like fold [76,77], 3CL^pro^ features the chymotrypsin-like fold [78,79].

## 3. Role of Viral Proteases in Innate Immune Antagonism

Beside their role in maturation of viral polyproteins, viral proteases cleave host proteins, particularly those involved in innate antiviral immunity (Figure 2), thereby neutralizing host defenses and creating an environment that favors virus replication. In this section, we will first provide a brief overview of cellular proteins that perform various functions in innate immune defenses and then present examples from the literature illustrating viral protease-mediated cleavage of these proteins.

### 3.1. Immune Sensors

#### 3.1.1. Overview

Innate immune signaling begins with the detection of PAMPs, either on the cell surface or within cellular compartments, through specialized host proteins, called pattern recognition receptors (PRRs) [80]. The PRRs involved in the recognition of positive-strand RNA viruses can be divided into two broad categories according to the cellular area they surveil. The first category consists of toll-like receptors (TLRs), which reside on the cell surface and in endosomes and sense extracellular and endosome-localized viral signatures [81,82]. The TLR family comprises 10 members (TLR1–10) in humans [83], of which TLR3, TLR7, and TLR8 have been implicated in detection of viral RNA. TLR3 recognizes double-stranded RNA, whereas TLR7 and 8 detect single-stranded RNA [84]. The second group of PRRs include RIG-I-like receptors (RLRs) [85] and nucleotide-binding oligomerization domain (NOD)-like receptors (NLRs) [86], which recognize viral nucleic acids in the cytoplasm of infected cells. Three major RLRs identified to date are RIG-I [87], MDA5 [88], and LGP2 [89], which all sense viral RNA, but exhibit differences in their recognition specificity and functional characteristics. Twenty-three NLR proteins have been described in humans [90,91]. Among these, NOD2 has been shown to recognize viral RNA and induce antiviral responses in infected cells [92].

**Figure 2 pathogens-11-00522-f002:**
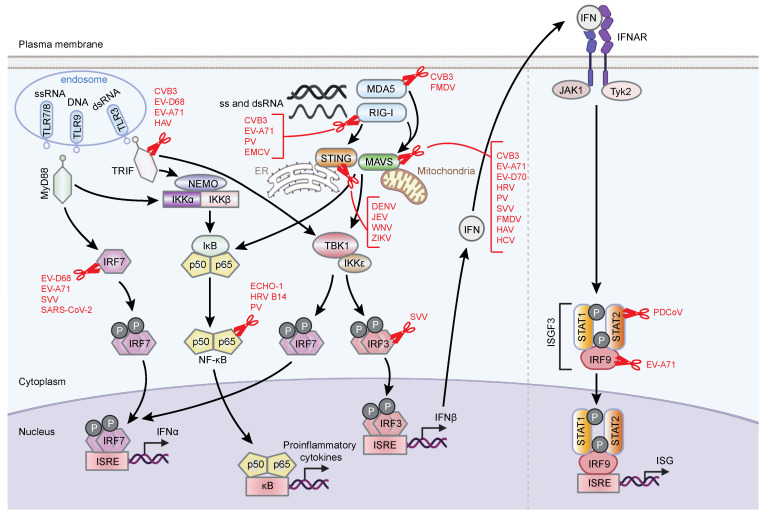
Antiviral innate immune pathways and their targeting by viral proteases. PAMPS associated with positive-strand RNA viruses are sensed by TLR3, TLR7/8, and TLR9 in endosomes and by RIG-I and MDA5 in the cytoplasm [93]. These sensors then interact with downstream adaptor proteins to initiate multiple signaling cascades. TLR7/8 and TLR9 employ MyD88 as an adaptor, whereas TLR3 signals through the TRIF adaptor. RIG-I and MDA5 use MAVS as an adaptor protein, although RIG-I has been reported to also signal through STING. The adaptor proteins, once bound by immune sensors, assemble large signaling complexes containing several cellular proteins, ultimately activating two major transcription factors, NF-κB and IRFs. These transcription factors travel to the nucleus and induce the expression of genes encoding proinflammatory cytokines, IFNs, and ISGs. IFNs are released into the extracellular space, where they bind to their respective cell surface receptors and trigger phosphorylation-dependent activation of pre-associated receptor tyrosine kinases, such as JAK1 and TYK2 [94]. This leads to recruitment and phosphorylation of STAT proteins. STAT1 and STAT2 form a heterodimer, which in turn recruits IRF9 to form the ISGF3 complex. This complex translocates to the nucleus and binds ISRE promoter elements, inducing the expression of ISGs. Viruses that employ their proteases to target various innate immune components are shown in red.

#### 3.1.2. Viral Cleavage of Immune Sensors

Viruses target PRRs to evade innate immune surveillance. Due to their localization on the cell surface and in endosomal compartments, TLRs mostly remain shielded from exposure to viral proteases. The RLRs, on the other hand, reside in the cytoplasm and are frequently targeted by viral proteases. The Coxsackievirus B3 (CVB3) 2A^pro^ and FMDV L^pro^ have been shown to cleave MDA5 [95,96]. The cleavage occurs at a highly conserved RGRAR motif and renders MDA5 non-functional by separating the viral RNA-binding C-terminal domain (CTD) from the two signal-transducing N-terminal caspase activation and recruitment domains (CARDs) that mediate interaction with the downstream adaptor proteins [96]. Notably, MDA5 is also bound by 3C^pro^ of several picornaviruses, such as Coxsackievirus-A6 (CV-A6), CV-A16, EV-D68, and EV-A71. However, this interaction does not involve MDA5 cleavage, but instead limits the downstream interaction of MDA5 with MAVS [28,97].

RIG-I is also targeted by viral proteases. The 3C^pro^ of EV-A71, CVB3, PV, and EMCV has been reported to inhibit RIG-I [95,98]. Conflicting data exist about the mechanisms that underlie 3C^pro^-mediated RIG-I inhibition. While Feng et al. reported that EV-A71 3C^pro^ cleaves RIG-I [95], Lei et al. showed that EV-A71 3C^pro^ acts by binding to N-terminal CARD domains of RIG-I and inhibiting the subsequent recruitment of MAVS [97]. These conflicting findings can be attributed to different experimental settings used in these studies; while Feng et al. examined the fate of RIG-I in virus-infected cells, Lei et al. performed their studies in cells overexpressing 3C^pro^. It is possible that in an overexpression setting, the viral protease does not achieve high enough localized concentration required for RIG-I cleavage. Alternatively, the 3CD protease precursor formed during the polyprotein processing in infected cells may be the main protease responsible for cleaving RIG-I. There are examples in the literature where the protease precursors of picornaviruses, and not the mature protease, cleave a cellular protein [99,100].

There is not much literature on LGP2 processing by viral proteases. The FMDV L^pro^ has been reported to cleave LGP2 [96]. As with MDA5, the protease targets the RGRAR sequence in the conserved helicase motif of LGP2, yielding protein products that can no longer regulate antiviral immunity in infected cells.

### 3.2. Adaptor Proteins

#### 3.2.1. Overview

Upon binding with PAMPs, the PRRs get activated and engage specific adaptor proteins, which serve as scaffolds for the assembly of large signaling complexes, called signalosomes. These signaling bodies then transduce signals to the downstream effector proteins. The TLRs signal via two major adapters, MyD88 and TRIF, leading to activation of transcription factors NF-κB and interferon-regulatory factors (IRFs) [101,102,103,104,105]. The MyD88-dependent pathway operates through phosphorylation-dependent degradation of the NF-κB inhibitory protein IκBα [104,106]. In unstimulated cells, NF-κB resides in the cell cytoplasm in an inhibitory complex with IκBα [107]. Activation of the MyD88 pathway and subsequent degradation of IκBα liberates NF-κB, which then migrates to the nucleus and induces the expression of proinflammatory genes. The TRIF-dependent pathway stimulates IRF3 and NF-κB signaling cascades, leading to expression of proinflammatory genes and type I interferon (IFN) [101,108,109].

The RLRs signal through the adaptor protein, MAVS (also known as IPS-1/VISA/Cardif) [110,111,112,113], that rapidly recruits a large number of signaling proteins to activate IRF and NF-κB pathways [111]. MAVS has three functional domains; an N-terminal CARD domain for interaction with immune sensors, a C-terminal transmembrane domain for localization to distinct cellular places, such as mitochondrial membranes, peroxisomes, and mitochondrial-associated membranes, and a proline-rich domain for engagement with functional partners [111,114,115,116,117,118]. Based on the composition of the MAVS signalosome, the RLR signaling bifurcates into two molecular cascades. The first cascade involves a protein complex of Tank binding kinase-1 (TBK1) and IκB kinase epsilon (IKKε), which directly phosphorylates IRF3 and IRF7 to induce the expression of IFN genes. The second cascade recruits the IKKα/β/γ complex (IKKγ is also known as NEMO) to cause phosphorylation and proteasomal degradation of IκBα, liberating NF-κB from the NF-κB- IκBα complex. The NF-κB then translocates to the nucleus and induces the expression of proinflammatory cytokines [119,120].

Another protein, stimulator of IFN gene (STING; also known as MITA, MPYS, and ERIS) [121,122,123,124], which mainly serves as an adaptor in the DNA sensing pathway, has been shown to participate in the transmission of RIG-I, but not MDA5, signals [122,125]. This role of STING is mediated by its direct interaction with RIG-I and MAVS in a complex that is stabilized upon virus infection, leading to activation of IRFs [122,123,125]. STING is also activated by the mitochondrial DNA being released into the cytoplasm of RNA virus-infected cells, which is then detected by the canonical DNA sensor, cGAS [126], triggering downstream activation of STING and subsequent production of IFN [127,128,129].

#### 3.2.2. Viral Cleavage of Adaptor Proteins

The adaptor proteins represent central signaling hubs of antiviral networks and are therefore an attractive target of virus-mediated proteolysis. MAVS is probably the most well characterized target of viral proteases. It is cleaved by a number of proteases from diverse virus families. Picornaviral proteases, including those of CVB3, PV, rhinoviruses, EV-D70, and EV-A71 [14,95,130], are known to cleave MAVS, with the cleavage site varying between viruses [14]. While these cleavages are mainly carried out by 2A^pro^ [95,131,132], L^pro^ and 3C^pro^ of some picornaviruses, such as CVB, Seneca Valley virus (SVV), and FMDV, have also demonstrated the ability to cleave MAVS [130,133,134]. Another picornavirus, HAV, employs a 3C^pro^ precursor, 3ABC^pro^, to cleave MAVS [100]. The transmembrane domain of 3A enables 3ABC to anchor on the mitochondrial membrane, and the protease activity of 3C^pro^ then carries out the MAVS cleavage. The mature 3C^pro^ alone does not localize to mitochondria and is therefore incapable of targeting MAVS. The HCV NS3-NS4A^pro^ also cleaves MAVS, leading to suppression of the innate immune pathway [112,135,136]. Mechanistically, in all cases of viral protease-mediated cleavage, the CARD domain of MAVS is dislodged from the mitochondrial membrane, thereby blocking formation of the MAVS signalosome and disrupting signal transduction.

TRIF is also a common target of viral proteases. Both 2A^pro^ and 3C^pro^ of picornaviruses have been reported to process TRIF [130,132,137,138]. However, like MAVS, TRIF also seems to be cleaved at different sites by different viruses; while the EV-71 3C^pro^ cleaves TRIF at only one site [138], the EV-D68 3C^pro^ cleaves at two sites [137]. It is currently unknown if these differential cleavage patterns are linked to differences in the biological response. Interestingly, HAV protease-polymerase precursor 3CD^pro^, but not the mature 3C^pro^, cleaves TRIF [99]. The TRIF cleavage has also been reported for HCV NS3-NS4A^pro^ [139].

Several positive-strand RNA viruses employ their proteases to target the STING protein, authenticating the relevance of this DNA sensing adapter to RNA viruses. Many flaviviruses, such as ZIKV, DENV, WNV, and JEV, but not YFV, use their NS2B-NS3^pro^ to cleave the human STING protein [140,141,142,143]. Notably, these viruses do not cleave mouse STING, and this deficiency has been linked, at least for ZIKV, to restricted viral replication in murine cells; knocking out the expression of STING in mouse cell lines caused ~10–50-fold increase in ZIKV replication [140]. Similarly, the DENV NS2B-NS3^pro^, although efficiently cleaves human STING, does not target the STING versions found in most other mammals, including non-human primates, possibly explaining the differential replication levels achieved by DENV in human versus other species [143].

### 3.3. Transcription Factors

#### 3.3.1. Overview

The signaling cascades unleashed upon sensing of viral PAMPs in infected cells lead to production of proinflammatory cytokines and IFNs. Two major transcription factors that regulate induction of these genes are NF-κB and IRFs [144,145]. The NF-κB family comprises five structurally related members, including NF-κB1 (also known as p50), NF-κB2 (also known as p52), RelA (also known as p65), RelB, and c-Rel [146]. These proteins, paired in distinct homo- and heterodimers, bind to a specific DNA element, κB enhancer, to mediate the transcription of target genes [147]. As described above, the NF-κB proteins normally exist in a latent form in unstimulated cells through binding to members of the IκB family of inhibitory proteins, mainly IκBα, which masks the nuclear localization signal of associated NF-κB proteins. Activation of immune signaling causes degradation of IκBα, triggered through its site-specific phosphorylation by a multi-subunit IκB kinase (IKK) complex [146,148]. IKK has two catalytic subunits, IKKα and IKKβ, and a regulatory subunit, NEMO (also called IKKγ) [149]. Once activated, IKK phosphorylates IκBα at two N-terminal serines and triggers ubiquitination-dependent IκBα degradation [107,148], allowing rapid and transient translocation of NF-κB members, predominantly the p50/RelA and p50/c-Rel dimers, to the nucleus, where they bind κB enhancers and induce the transcription of hundreds of target genes, most of which encode proinflammatory cytokines [107,146,148,150,151,152,153].

The second group of transcription factors are IRFs that are primarily involved in transcription of IFN and IFN-stimulated genes (ISGs). Since the discovery of the first IRF, IRF1, in 1988 [154,155], the total of nine IRFs (IRF1–9) have been characterized in mammals. They bind to a specific DNA motif, called IFN-stimulated response element (ISRE), as distinct homo- and hetero-dimers and regulate the expression of a large cohort of genes [156]. Apart from interacting among themselves, IRFs also partner with other transcription factors, such as STATs and NF-κB, activating a broad spectrum of genes and controlling diverse transcriptional programs [157]. While all IRFs play roles in immune response of cells, IRF3 and IRF7 are the crucial modulators of IFN production [158,159,160]. Both IRF3 and IRF7 exist in an inactive form in the cytoplasm of uninfected cells [161,162]. Upon virus infection, they are activated through TBK1 and IKKε-mediated phosphorylation [163,164,165,166], which induces conformational changes in IRFs, facilitating their dimerization and subsequent nuclear translocation, where, along with other co-activators, they regulate transcription of type I IFN and ISGs [1,167].

#### 3.3.2. Viral Cleavage of Transcription Factors

IRFs and NF-κB are both targeted by viral proteases. EV-A71 and EV-D68 employ their 3C^pro^ to process IRF7 [168,169], the main transcription factor involved in IFNα production. While EV-A71 3C^pro^ cleaves IRF7 only at one site [168], EV-D68 3C^pro^ processes the protein at two sites [169]; however, the functional consequences of these differential cleavages are unknown. Another picornavirus, SVV, has been reported to reduce cellular levels of both IRF3 and IRF7 in a manner that depends on the enzymatic activity of the viral 3C^pro^ [170]. A recently published study demonstrated the ability of the SARS-CoV-2 PL^pro^ to cleave IRF3 [171]. The NF-κB family members are also proteolytically processed by viral proteases. The 3C^pro^ of PV, ECHO-1, and human rhinovirus B-14 cleaves the p65 subunit of the NF-κB complex, thereby shutting down the induction of proinflammatory genes [172]. In addition to direct cleavage, viral proteases can also block the nuclear migration of transcription factors by targeting the protein transport machinery. In this vein, HCV NS3-NS4A^pro^ has been shown to inhibit IFN production by cleaving importin β1, a key nucleocytoplasmic transport receptor involved in nuclear import of IRF3 and NF-κB [173].

### 3.4. The IFN Response Pathway

#### 3.4.1. Overview

The IFN family includes three main classes of related cytokines: Type I IFNs, type II IFN, and type III IFNs. Once released from virus-infected cells, they work in an autocrine and paracrine manner to create an antiviral state in both infected cells and the neighboring bystander cells. Interaction of IFNs with their receptors triggers activation of receptor tyrosine kinases JAK1 and TYK2, which in turn phosphorylate members of the STAT family of proteins, triggering their dimerization [174]. STAT1 and STAT2 are the main IFN-activated transcription factors, which, together with IRF9, form a trimeric complex, ISGF3, that travels to the nucleus and drives transcription of ISGs [175]. These ISGs, which are several hundred in number, are the workhorse of the innate immune system and employ a variety of mechanisms to block and/or eliminate viral infections [6,7].

#### 3.4.2. Ablation of IFN Responses by Viral Proteases

In addition to ablating the IFN production pathway, viral proteases also block events that happen after IFNs bind to their receptors on the cell surface. In an elegant study, Morrison et al. showed that several picornaviruses, such as EV-D70, HRV A16, and PV, are capable of replicating in type I IFN-treated cells in a 2A^pro^-dependent manner [176]. While mechanisms through which 2A^pro^ limits the antiviral effect of IFN are unknown, the authors speculated that the protease might be cleaving ISGs responsible for inhibition of picornavirus replication. In another study, the EV-A71 3C^pro^ was shown to target IRF9 for proteolytic cleavage [177], abrogating the ability of IFN to inhibit viral replication. Porcine deltacoronavirus (PDCoV) has been reported to employ its 3CL^pro^ to cleave STAT2 and inhibit the ISRE reporter activity [178].

## 4. Concluding Remarks

This review summarized the role of the viral protease activity in counteracting host defenses. Historically, the study of virus-mediated proteolysis of host proteins has relied on targeted molecular biology techniques, such as western blot. However, with terrific advances in systems biology approaches, it has now become possible to perform a global survey of protein cleavages in virus-infected cells [14,179]. Defining the full extent of proteolysis in virus-infected cells and performing the functional characterization of newly identified host substrates can illuminate the complexities of the innate immune system and highlight the role of viral proteases as important virulence factors. Further, most published studies have only utilized mature forms of viral proteases to test the cleavage of a cellular protein, running the risk of yielding false-negative results. There are examples when the precursor forms of a viral protease, but not its mature form, cleaves a host protein [99,100]. This could be due to altered subcellular localization of the precursor form or changes in the contours of the protease active site, modulating the affinity of substrate binding and thereby influencing the substrate cleavage efficiency or specificity. Therefore, unless a protein is tested in virus-infected cells, the possibility of it being cleaved by a viral protease should not be ruled out. Moreover, in some cases, only a small fraction of a given cellular protein is being targeted for cleavage. For example, while the cleavage of MAVS in picornavirus-infected cells disrupts antiviral signaling, the total amount of full-length MAVS in virus-infected cells appears to remain relatively unchanged when tested by western blot [14]. This can be attributed to a possibility that, at least in some cases, viral proteases target only the portion of a host protein localized to a particular subcellular organelle. For instance, it is conceivable that only the fraction of MAVS localized to mitochondrial membranes or peroxisomes is being targeted by a viral protease. With rapidly improving performance of proteomic approaches and increasing interest in viral proteases as attractive drug targets, it should soon become possible to define precise spatiotemporal regulation of protein cleavages in virus-infected cells.

## Figures and Tables

**Figure 1 pathogens-11-00522-f001:**
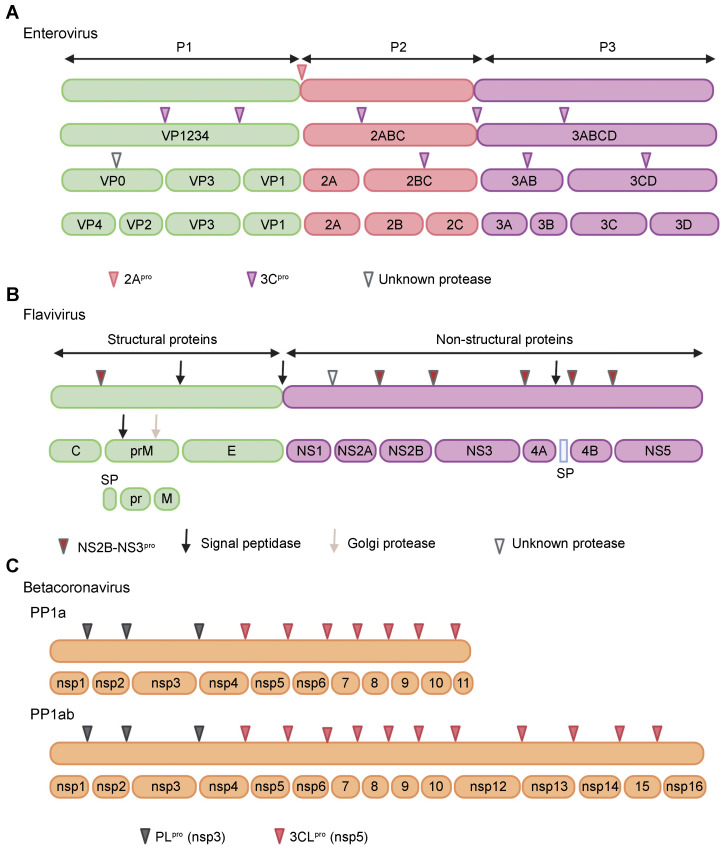
The polyprotein processing of a representative genus from the *Picornaviridae*, *Flaviviridae*, and *Coronaviridae* families. (**A**) The polyprotein of the Enterovirus genus of the *Picornaviridae* family consists of three regions, P1, P2, and P3. P1 contains structural proteins, whereas P2 and P3 contain non-structural proteins. The first cleavage in the polyprotein is mediated by 2A^pro^, which cleaves at its N-terminus, separating P1 from P2-P3. P1 is further processed by 3C^pro^ to yield three proteins, VP0, VP3, and VP1. Very late in infection, VP0 is cleaved into VP2 and VP4 by an unknown protease, although some studies suggest that it is an autocatalytic reaction [46,47]. P2 and P3 are cleaved by 3C^pro^ to liberate the total of seven non-structural proteins. (**B**) The polyprotein of the Flavivirus genus of the *Flaviviridae* family has two regions: One containing structural proteins and the other containing non-structural proteins. These proteins are liberated from the polyprotein through the action of host and viral proteases. Flaviviruses encode only one protease, NS2B-NS3^pro^, that mediates six cleavages in the polyprotein. SP, signal peptide (**C**) Members of the Betacoronavirus genus of the *Coronaviridae* family encode two overlapping polyproteins PP1a and PP1ab. PP1ab results from a (−1) ribosomal frameshift overreading the stop codon of ORF1a, leading to identical N-terminal ends of PP1a and PP1ab and a long C-terminal extension in PP1ab. Both polyproteins are processed by two viral proteases, PL^pro^ (encoded by the nsp3 gene) and 3CL^pro^ (a.k.a. M^pro^; encoded by the nsp5 gene) to liberate a total of 16 proteins.

## Data Availability

Not applicable.

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
