# Peer review of "Surgical Strikes on Host Defenses: Role of the Viral Protease Activity in Innate Immune Antagonism"

_pathogens, 2022, doi:10.3390/pathogens11050522_

Round 1

Reviewer 1 Report

Chin and Saeed have written an excellent review on the topic of viral proteases and their roles in antagonizing the innate immune response. Overall, this review is well-written and provides a good overview of prior literature on viral proteases and innate immunity. Below are several minor edits:

1) At least two other thorough reviews (PMID 33692152 and 33624382) have recently been written on viral proteases in the last year or so. Though more focused on specific viruses, such as picornaviruses and coronaviruses, readers of this review would benefit from Chin and Saeed adding in references to those reviews. 

2) The authors' description of the role of STING in the innate immune response (lines 222-226) is a bit incomplete. While the authors cite literature suggesting STING interacts with RIG-I and MAVS, another established role of STING in the innate immune response to RNA viruses comes from its signaling role downstream of cGAS sensing DNA that has been released from mitochondria during RNA virus infection. The authors should mention this as a potential mechanism for STING's role against RNA viruses.

3) There are a few places in which the review reads as though it is less focused on viral proteases and more as a listing all of the various innate immune pathways and interactions that exist. In a review that already contains many acronyms and pathways, it would be useful to streamline the description to eliminate some of this information and keep it focused on targets of viral proteases. Two examples include lines 211-214, which lists a number of interactors with MAVS that are not mentioned elsewhere in the review and are not targeted by proteases, and lines 307-312, which describe the various subtypes of interferons that are again not discussed elsewhere. Any chance the authors have to eliminate such information will make this review more accessible to readers who are not extremely familiar with the innate immune system.

Author Response

Dear Reviewer

Thank you very much for the insightful comments! Below is the point-by-point response to your suggestions.

1) At least two other thorough reviews (PMID 33692152 and 33624382) have recently been written on viral proteases in the last year or so. Though more focused on specific viruses, such as picornaviruses and coronaviruses, readers of this review would benefit from Chin and Saeed adding in references to those reviews. 

The suggested references have been added to the text.  

2) The authors' description of the role of STING in the innate immune response (lines 222-226) is a bit incomplete. While the authors cite literature suggesting STING interacts with RIG-I and MAVS, another established role of STING in the innate immune response to RNA viruses comes from its signaling role downstream of cGAS sensing DNA that has been released from mitochondria during RNA virus infection. The authors should mention this as a potential mechanism for STING's role against RNA viruses.

Good point. This concept has been incorporated to the section.  

3) There are a few places in which the review reads as though it is less focused on viral proteases and more as a listing all of the various innate immune pathways and interactions that exist. In a review that already contains many acronyms and pathways, it would be useful to streamline the description to eliminate some of this information and keep it focused on targets of viral proteases. Two examples include lines 211-214, which lists a number of interactors with MAVS that are not mentioned elsewhere in the review and are not targeted by proteases, and lines 307-312, which describe the various subtypes of interferons that are again not discussed elsewhere. Any chance the authors have to eliminate such information will make this review more accessible to readers who are not extremely familiar with the innate immune system.

We agree with the reviewer that the information provided on lines 211-214 and 307-312 was mostly frivolous in the context of this review. Therefore, we have removed this information from the text.

Reviewer 2 Report

The review by Chin and Saeed is an overview of the role of the proteases encoded by three families of RNA viruses in counteracting innate immunity.
Chpaters 1 and 2 are fine in my view, just a little consideration about figure1:while the authors choose to analyze three viral families, starting by picornaviridae, fig. 1 depicts a specific genus, Enteroviruses along
with the other two families. It would be better to use three generic schemes to show the three families, or one representative genus for each.
Chapter 3 is merely a short introduction to chapters 4 and following. I suggest to downgrade chapters 4-7 to 3.1 to 3.4, and introduce sub-paragraphs where
really necessary. These chapters are anyways not so easy to follow, and would benefit from having descriptive figures and summarizing tables (i.e, fig2 could remain as general
overview of the innate immuniti factors, but the action of proteases on specific elements of the pathway could be depicted in a focus figure for each chapter).

Some minor points:
line 54-60: references 33-34 are missing
line 145: please add one or more appropriate references in fig2 description
line 247: ref 141 appears in brackets
line 260-264: please add a reference
line 297: ref 170 appears in brackets
line 320: please add a reference
line 426: please correct reference style
line 470: please correct reference style
line 490: please correct reference style

Author Response

Dear Reviewer

Thank you very much for the insightful comments! Below is the point-by-point response to your suggestions.

Chapters 1 and 2 are fine in my view, just a little consideration about figure1: while the authors choose to analyze three viral families, starting by picornaviridae, fig. 1 depicts a specific genus, Enteroviruses along with the other two families. It would be better to use three generic schemes to show the three families, or one representative genus for each.

Excellent suggestion. We have now included one representative genus for each of the three virus families. The figure and the associated description have been changed accordingly.

Chapter 3 is merely a short introduction to chapters 4 and following. I suggest to downgrade chapters 4-7 to 3.1 to 3.4, and introduce sub-paragraphs where really necessary. These chapters are anyways not so easy to follow, and would benefit from having descriptive figures and summarizing tables (i.e, fig2 could remain as general
overview of the innate immunity factors, but the action of proteases on specific elements of the pathway could be depicted in a focus figure for each chapter).

We have changed the chapter numbering as per reviewer’s suggestion. However, we have decided to stick with the original figure. In our opinion, having a separate illustration for each subheading will dilute the message and make it challenging to provide a holistic overview of the extent to which viral proteases alter various steps in the immune pathway.  

Some minor points:

line 54-60: references 33-34 are missing

Added

line 145: please add one or more appropriate references in fig2 description

Added

line 247: ref 141 appears in brackets

We added brackets to separate the reference from the superscripted text.

line 260-264: please add a reference

Added

line 297: ref 170 appears in brackets

We added brackets to separate the reference from the superscripted text.

line 320: please add a reference

Added

line 426: please correct reference style

Corrected

line 470: please correct reference style

Corrected

line 490: please correct reference style

Corrected

Reviewer 3 Report

Overall, the review paper written by Chin and Saeed is well organized and it gives the reader a good overview of innate immunity mechanisms targeting positive stand RNA viruses. 

Section 1   

Line 22 to 24 : Pathogen recognition receptors such as TLR or RLR almost exclusively need an adaptor protein to relay signalling to downstream effector molecules. It would be useful to add this concept for the reader.   

Line 31: Viruses need to circumvent host defences to do more than replication: they need to act at each step of their life cycle which includes entry, replication, assembly and egress. It would be useful to nuance this statement.   

Section 2 

Line 49 to 117: Very descriptive. A table with key take away would be nice in a review: ex. Family, names of the virus, genome size, key viral proteins with protease activity, know antiviral/pro-viral role.   

Section 3 

Line 131-133: TLRs senses extracellular and intracellular RNA and not ‘’ actively engulfed pathogens’’. Please rephrase.   

Line 133: It would be nice to add what are the ligands of all the TLRs that you are including in your list. Especially since, in the introduction, you are talking about proteins as PAMPs. Otherwise, you should restructure the paragraph to state that you are focusing on RNA sensors.  

Line 139-140: You should add the specificity of RNA ligands for RIG-I and MDA5. Ie. 5’-ppp. General comment: there is a lot of detailed signalling events that are talked about in this section, but general concepts are not introduced elsewhere (ex. CARD domains). It might be a good idea to add a general description of the RLR signalling pathway with highlights connected to the text. Information might be elsewhere in the text (Ex. In the next section) but it is more educational to build upon concepts that have been clearly presented before going into the details.  

Line 223-226: It is not clear if you consider STING an obligatory or a dispensable adaptor protein for the RIG-MAVS-IRF3 signalosome. Is its necessary protection against all RNA viruses? A lot of data exists on this subject. It would be a great addition to this section.   

Line 229 to 240: A think that it would be nice to differentiate why MAVS is not able to serve as an adaptor protein. It is because it is dislodged from the mitochondria or because it cannot interact with RIG-I or IRF-3 or because it cannot form MAVS-aggregates? A lot of data exists on this subject, adding depth to the text would be useful for the reader.   

Section 6   

Another mechanism to prevent IRF3 translocation has been described such as cleavage of importin beta by HCV NS3/4A. Might be useful to add the description of other ways that viral proteases are able to counteract transcription factors activation or translocation.

Author Response

Dear Reviewer

Thank you very much for the insightful comments! Below is the point-by-point response to your suggestions.

Section 1   

Line 22 to 24 : Pathogen recognition receptors such as TLR or RLR almost exclusively need an adaptor protein to relay signalling to downstream effector molecules. It would be useful to add this concept for the reader.  

In this opening paragraph of the review, we only provided a general concept of the innate immune pathway. Various pathogen recognition receptors and their associated adapter proteins have been introduced in the subsequent sections. Therefore, we respectfully disagree with the reviewer and would prefer to leave this part as is.  

Line 31: Viruses need to circumvent host defences to do more than replication: they need to act at each step of their life cycle which includes entry, replication, assembly and egress. It would be useful to nuance this statement.   

Good point! We have replaced “replication” with “lifecycle”.

Section 2 

Line 49 to 117: Very descriptive. A table with key take away would be nice in a review: ex. Family, names of the virus, genome size, key viral proteins with protease activity, know antiviral/pro-viral role.   

Several published review articles on this topic, which we have cited in our manuscript, contain such tables. Therefore, we decided against incorporating a table. However, we have now simplified the description of viral families by rearranging pieces of information.

Section 3 

Line 131-133: TLRs senses extracellular and intracellular RNA and not ‘’ actively engulfed pathogens’’. Please rephrase.   

Thanks for pointing this out! We have replaced “actively engulfed pathogens” with “endosome-localized viral signatures”.

Line 133: It would be nice to add what are the ligands of all the TLRs that you are including in your list. Especially since, in the introduction, you are talking about proteins as PAMPs. Otherwise, you should restructure the paragraph to state that you are focusing on RNA sensors.  

To simplify this part, we have now removed the mention of all TLRs and instead only described TLRs implicated in detection of viral RNA.

Line 139-140: You should add the specificity of RNA ligands for RIG-I and MDA5. Ie. 5’-ppp. General comment: there is a lot of detailed signalling events that are talked about in this section, but general concepts are not introduced elsewhere (ex. CARD domains). It might be a good idea to add a general description of the RLR signalling pathway with highlights connected to the text. Information might be elsewhere in the text (Ex. In the next section) but it is more educational to build upon concepts that have been clearly presented before going into the details.  

We have now added the description of CARD domains in section 3.1.2, relating the cleavage site targeted by viral proteases in immune sensors.

Line 223-226: It is not clear if you consider STING an obligatory or a dispensable adaptor protein for the RIG-MAVS-IRF3 signalosome. Is its necessary protection against all RNA viruses? A lot of data exists on this subject. It would be a great addition to this section.   

We have added information about the role of STING in eliciting innate immunity against positive-strand RNA viruses.

Line 229 to 240: A think that it would be nice to differentiate why MAVS is not able to serve as an adaptor protein. It is because it is dislodged from the mitochondria or because it cannot interact with RIG-I or IRF-3 or because it cannot form MAVS-aggregates? A lot of data exists on this subject, adding depth to the text would be useful for the reader. 

Excellent point. We have now added this information to the text.   

Section 6   

Another mechanism to prevent IRF3 translocation has been described such as cleavage of importin beta by HCV NS3/4A. Might be useful to add the description of other ways that viral proteases are able to counteract transcription factors activation or translocation.

We thank the reviewer for this valuable suggestion. This concept has now been included in the text.